# Physical Health Among Adults with Depressive Symptoms in Qatar: Findings from Qatar Biobank Population-Based Study

**DOI:** 10.3390/ijerph22050726

**Published:** 2025-05-02

**Authors:** Mohammed Aldalaykeh, Ahmad H. Abu Raddaha, Fadwa Alhalaiqa, Waqas Sami, Fadi Khraim

**Affiliations:** College of Nursing, QU Health Sector, Qatar University, Doha P.O. Box 2713, Qatar; a.aburaddaha@qu.edu.qa (A.H.A.R.); f.alhalaiqa@qu.edu.qa (F.A.); waqas@qu.edu.qa (W.S.); fkhraim@qu.edu.qa (F.K.)

**Keywords:** physical health, depression symptoms, body mass index, heart rate, hypertension

## Abstract

Depression is a mood disorder that is highly prevalent and is characterized by cognitive, emotional, and physical symptoms. This study aimed to describe the physical health status of individuals with depressive symptoms in Qatar and examine the relationships between physical health indicators and depressive symptoms. A cross-sectional correlational design was used, and data were extracted from the Qatar Biobank. Depression was measured using the Patient Health Questionnaire (PHQ-9). The physical health measurements included heart rate, blood pressure, lung capacity, body mass index (BMI), waist-to-hip ratio, and a self-reported questionnaire. The sample included 687 participants with an average age of 35.39 ± 9.82 years old. The comorbidities reported by participants included diabetes, hypertension, and hypercholesterolemia, and their prevalence ranged from 10 to 26.2%. The BMI data revealed that 38.6% of the participants were either overweight or obese. Approximately 29% of females and 22% of males had an abnormally high waist-to-hip ratio. The percentage of individuals who complained of recurrent chest pain, headache/migraine, or body pain ranged between 12.2 and 43.2%. The mean score of the PHQ-9 was 10.07 ± 4.05, and 43.1% of the sample reported having moderate to severe depression. Several factors were significantly related to depression scores: heart rate, waist-to-hip ratio, headache/migraine, chest pain, body pain, and recent weight change. Healthcare professionals should screen individuals for both depressive symptoms and physical symptoms. This may help in the earlier detection of mental illness and/or physical disease, and thus may ensure better recovery or prognosis and may prevent serious complications.

## 1. Introduction

Health is traditionally conceptualized within the biopsychosocial model, which integrates the physical, psychological, and social dimensions of well-being. This holistic framework recognizes the intricate connections between mental health and physical health, positioning mental well-being as a crucial component of overall health [1,2]. Depression is a mood disorder that is characterized by a continuous sad mood and a lack of pleasure or interest in performing activities or tasks that could be used to increase happiness for the individual [3]. According to the Diagnostic and statistical manual of mental disorders (DSM-5-TR), Major Depressive Disorder is diagnosed based on the presence of at least five symptoms over a two-week period, including a depressed mood, anhedonia, weight changes, sleep disturbances, psychomotor changes, fatigue, feelings of worthlessness or guilt, concentration difficulties, and suicidal ideation [4]. These criteria offer a clinically robust framework for understanding the manifestation of depression in the adult population.

According to the World Health Organization, approximately 5% of adults are diagnosed with depression globally, and this rate is higher among females and elderly people [5]. The prevalence of depression in Qatar is similar to the global rate and ranges between 4.2 and 6.6% [6]. In the context of adulthood, depressive symptoms are influenced by both psychosocial stressors and physiological changes. Adults often face increased responsibilities, career pressures, family dynamics, and emerging or worsening chronic conditions, all of which may contribute to the onset or exacerbation of depression [1,2,3].

The diagnostic criteria for major depression include several physical clinical manifestations such as weight loss or weight gain, fatigue, and insomnia or hypersomnia [4]. This highlights the association between physical health and mental health, which is consistent with the findings of previous studies. For example, 15.4–20.1% of type 2 diabetic patients have depression [7,8], and this rate is significantly higher among patients with end-stage renal disease, reaching 48% [9]. Additionally, several studies have highlighted the relationship between depression and hypertension [10,11,12]. It is important to mention that the relationship between depression and physical diseases is two-way in nature [3,13,14]. This means that physical diseases may affect the level of one’s energy, hormones, neurotransmitters, and psychosocial functioning, which may lead to depressive symptoms [3,13,14]. On the other hand, depression may cause an abnormality in hormones or neurotransmitters and can affect self-care behaviors and nutritional habits (e.g., loss of appetite), which may cause nutritional deficiencies and electrolyte imbalances, which ultimately may lead to physical diseases [3,13,14].

Physical health can be assessed subjectively through self-reported questions and surveys, and it can be assessed objectively via many indicators including (but not limited to) anthropometrics, vital signs, blood tests, spirometry, electrocardiograms, and body imaging [15]. Several studies have examined the relationships between depressive symptoms and physical health indicators and symptoms; most of these studies have reported significant relationships. More specifically, indicators include body mass index (BMI), waist-to-hip ratio, blood pressure, lung capacity measured by Forced Expiratory volume (FEV1) and Forced Vital Capacity (FVC), heart rate, and lipid profile [11,12,16,17,18,19,20]. Also, several systematic reviews and meta-analysis studies have explored the relationship between depression and comorbidities or chronic illnesses such as cardiovascular diseases, diabetes mellitus, chronic obstructive pulmonary disease, stroke, arthritis, cancer, and Parkinson’s disease [21,22,23,24,25]. They reported consistent results of there being a significant relationship between depression and chronic physical illnesses and comorbidities. This may confirm the strong association between all aspects of health [21,22,23,24,25].

Many studies have investigated the relationship between depression and physical diseases or physical signs and symptoms, but these studies have not taken a comprehensive approach to examining several physical health indicators all at once, or their relationships with depression. Instead, each study depended on one or two physical health indicators and examined their relationship with depression. Considering the multifactorial etiology of depression, and to have a holistic and comprehensive approach when describing the physical health status of patients with depressive symptoms, assessing physical health via multiple health indicators that cover both subjective and objective data are recommended. Also, this study is important as it addresses a notable gap in the literature: the relationship between depressive symptoms and physical health among adults in non-Western contexts. Most existing studies have focused on Western populations, and this paper aims to expand the global understanding of mental–physical health interactions by analyzing a sample from Qatar.

This study aims to describe the physical health status of individuals with depressive symptoms in Qatar. Additionally, this study aims to examine the relationships between physical health indicators and depressive symptoms. The study hypothesizes that poorer physical health indicators (e.g., higher BMI, elevated blood pressure, and abnormal cholesterol levels) are associated with greater depressive symptoms among adults in Qatar.

## 2. Materials and Methods

### 2.1. Study Design and Sample

This was a cross-sectional correlational study that was based on a secondary analysis of data extracted from the Qatar Biobank (QBB). In 2012, a population-based cohort study was conducted by the QBB with the goal of collecting comprehensive health-related data from people living in Qatar, including both Qatari and long-term residents of Qatar (more than 10 years) [26]. Currently, the QBB has collected data from more than 24,000 participants. The data collection of the surveys was performed by expert researchers and trained research assistants, while the biological samples and lab tests were performed by healthcare professionals (e.g., physicians and nurses). Data collection was initiated at several public places in Qatar such as public parks and malls, and those who met the criteria and consented to participate were asked to complete the self-reported surveys and were informed about the date and location of the setting where biological samples, anthropometric measurements, vital signs, and lab tests would be collected [26].

This was a secondary analysis, so we relied on the QBB sampling method, which was a convenience sampling method. However, after submitting our proposal and our request to the QBB (because the data are not publicly accessible), the QBB provided us with the data of randomly selected participants who met the inclusion/exclusion criteria mentioned below. G*Power software (version 3.1.9.7) was used to calculate the sample size [27]. Multiple regression was used to guide the calculations, and we entered a small effect size (0.02), a significance level of 0.05, a power of 0.8, and a number of predictors of 6. The required sample size based on these criteria was 687.

Participants were included in this study if they were 18–55 years old, had a PHQ-9 score equal to five or higher (i.e., had at least mild depressive symptoms), and had been living in Qatar for at least 3 years. Participants were excluded from the study if they had any type of fracture or history of renal failure, heart failure, or COPD (because it may have negatively affected their lifestyle and mental health).

### 2.2. Depressive Symptoms

Depressive symptoms were measured using the Patient Health Questionnaire-9 (PHQ-9). This tool includes nine items. Each item is rated from zero (not at all) to three (nearly every day) [28]. The total score is calculated by summing the scores of the nine items. The total score ranges from 0 to 27, with higher scores indicating more severe depressive symptoms [28]. The total score can be classified into five categories: no depression (PHQ-9 score less than 5), mild depression (PHQ-9 score between 5 and 9), moderate depression (PHQ-9 score between 10 and 14), moderate-to-severe depression (PHQ-9 score between 15 and 19), and severe depression (PHQ-9 score between 20 and 27) [28]. The English and Arabic versions were used in this study, and the appropriate version was introduced to participants based on their language. The Arabic version was rigorously developed and tested in previous studies using forward and backward translation and expert review to ensure its cultural and linguistic adaptation [29,30]. The PHQ-9 is widely used to screen for depression and has shown very good to excellent reliability scores in previous studies [31,32,33,34,35]. In the current study, the PHQ-9 showed acceptable reliability scores (Cronbach’s alpha = 0.70).

### 2.3. Sociodemographic Factors and Physical Health Indicators

Anthropometric measurements (e.g., BMI and waist-to-hip ratio) were collected using a Seca Stadiometer. Pneumotrac Vitalograph from Vitalograph Ltd. (Ennis, County Clare, Ireland) was used to assess the respiratory function of the participants. Demographic characteristics, past health history, comorbidities, current health history, and lifestyle data were collected via a self-report survey that was developed by the QBB. Physical health examination, vital signs, and health history data were collected by nurses and clinicians at the QBB facility.

### 2.4. Statistical Analysis

Data management was performed before the actual analysis, which included screening for missing values, errors/typos, and outliers. Descriptive statistics (frequency, mean, and range) were used to describe the sample characteristics and major variables of the study (depressive symptoms and physical health status). The inferential statistics that were used to examine the relationship between physical health and depressive symptoms included an independent *t*-test, Pearson Correlation, Analysis of Variance (ANOVA), and multiple linear regression. For the ANOVA, the Bonferroni correction method was applied during post hoc analysis. Also, the Bonferroni correction method was applied during the multiple *t*-test analysis for the relationship between physical symptoms and the PHQ-9 scores to control for Type-I errors. In this case, alpha was adjusted to *p* < 0.0125 since we had four series of *t*-test analyses.

## 3. Results

The sample included 687 participants, with an average age of 35.39 ± 9.82 years, ranging from 18 to 55 years. Approximately 56% of the participants were female, 53% reported having a bachelor’s degree, and 65% reported having paid employment. Only 10% of the participants reported being diagnosed with diabetes, while 12.5% were diagnosed with hypertension, and 26.2% of the participants had hypercholesterolemia. See Table 1.

The physical health assessment revealed that the mean heart rate score was 72.09 ± 12.94, whereas the mean systolic and diastolic blood pressure scores were 111.62 ± 13.10 and 70.34 ± 10.37, respectively. Eighty-one (11.8%) participants had either bradycardia or tachycardia. On the other hand, 59 (8.6%) participants had a systolic blood pressure higher than 130, and 46 (6.7%) participants had a diastolic blood pressure higher than 85. The mean BMI score was 28.81 ± 6.41, and 265 (38.6%) participants were either overweight or obese, whereas only 25 (2.6%) participants were underweight. The mean score of the waist-to-hip ratio was 0.82 ± 0.10, and considering gender variation, 111 (28.8%) females had a waist-to-hip ratio greater than 0.80, while 66 (21.9%) males had a waist-to-hip ratio greater than 0.95. In terms of lung capacity, the mean FEV1/FVC score was 0.79 ± 0.10, and 73 (10.6%) participants had a ratio less than 0.70. See Table 2.

When participants were asked about their health during the previous year, 19.1% reported having a wheezing chest, and 43.2% reported having frequent headaches or migraines. Also, 12.2% of the participants reported having pain in several areas of the body, whereas 42.0% of the participants reported having chest pain during the previous year. Approximately 48% of the participants reported weight gain during the previous year, whereas 28% of the participants reported weight loss. Additionally, the participants were asked to describe their current health status. The findings revealed that 71.5% of the participants rated their health status as either excellent or good, whereas 28.5% of the participants rated their health status as either fair or poor. See Table 3.

The mean score of the PHQ-9 was 10.07 ± 4.05. A total of 391 (56.9%) participants reported having mild depression, whereas 208 (30.3%) participants had moderate depression. On the other hand, 68 (9.9%) participants had moderate-to-severe depression, and 20 (2.9%) participants reported having severe depression.

After the careful examination of the relationships between the PHQ-9 scores and the demographic characteristics of the participants and their physical health, the findings indicated that there was no significant relationship between the demographic variables, the history of chronic physical illnesses, and the PHQ-9 scores (see Table 1). To ensure that comorbidity and chronic physical illnesses were not a confounding variable to the relationship between physical symptoms/physical health indicators and the PHQ-9 scores, we ran an independent *t*-test analysis between participants without a history of chronic illnesses (*n* = 439) and those with chronic illnesses (*n* = 248). The results showed no significant differences in the PHQ-9 scores between the two groups (t = 0.148, *p* = 0.882). Also, to ensure that the number of comorbidities was not a confounding variable, we ran an ANOVA between three groups (no chronic illnesses vs. one chronic illness vs. two or more chronic illnesses) in terms of the PHQ-9 scores. The results showed no significant differences in the PHQ-9 scores between the three groups (F = 0.916, *p* = 0.401). This confirmed that chronic illnesses and the number of commodities were not confounding variables to the PHQ-9 scores, and thus there was no need to adjust the findings when examining the factors that affected the PHQ-9 scores, such as physical symptoms and physical health indicators.

There were several significant relationships between the PHQ-9 scores and physical health indicators. For example, Pearson Correlation results showed that there was a significant positive relationship between heart rate and PHQ-9 scores, whereas there was a significant negative relationship with the waist-to-hip ratio (see Table 2).

There was a significant relationship between self-reported physical signs and symptoms and PHQ-9 scores. The findings of the independent *t*-test revealed that participants who reported having recurrent headaches/migraines, body pain, and chest pain in the previous year had significantly higher PHQ-9 mean scores than those who did not experience these signs and symptoms (see Table 3). Also, the ANOVA findings showed that depressive severity was significantly associated with weight change during the previous year (F = 4.57, *p* < 0.05). Post hoc analysis showed that participants who gained weight during the last year showed significantly higher PHQ-9 scores than those who reported no change in their weight (*p* < 0.017, after Bonferroni correction). The remaining paired comparisons showed no significant results after Bonferroni corrections (see Table 3). Finally, the findings of the ANOVA revealed a significant relationship between the overall rating of health and the PHQ-9 scores (F = 29.69, *p* < 0.001), with the lowest PHQ-9 mean scores reported by those who rated their overall health as excellent, and the highest mean scores reported among those who rated their overall health as poor (see Table 3). Post hoc analysis showed that all paired comparisons (except the excellent–good comparison) were significant in terms of the PHQ-9 mean scores (*p* < 0.0125, after Bonferroni correction).

Simultaneous multiple linear regression analysis was performed to predict the PHQ-9 scores. After only selecting the factors that showed a significant relationship with the PHQ-9, the regression model included six predictors: heart rate, waist-to-hip ratio, headache/migraine, chest pain, body pain, and weight change. Dummy coding was performed for “weight change” before running the regression analysis. The regression model was significant (F = 7.34, *p* < 0.001), and it accounted for 7.2% of the explained variance in the PHQ-9 scores (Table 4). Four predictors (body pain, chest pain, heart rate, and recent weight change) had significant independent effects, whereas the waist-to-hip ratio and headache/migraine status were not significant predictors. The strongest predictor was body pain (β = 0.199, *p* < 0.001), followed by weight change (β = 0.117, *p* < 0.05). See Table 4.

## 4. Discussion

This study examined the physical health among individuals who were experiencing depressive symptoms. The findings showed that the prevalence of diabetes and hypertension among individuals with depressive symptoms ranged between 10% and 12.5%. The diabetes rate was consistent with the diabetes rate among the general population of Qatar (15.1%) and the Middle Eastern and North Africa region, which is estimated to be 9.3% [36]. On the other hand, the sample showed significantly lower rates of hypertension than the general Qatari population, which is estimated to range from 33 to 41% [37,38,39]. This was supported by the low percentage of people in this study with elevated systolic and diastolic blood pressure. A possible reason for the lower rates of diabetes and hypertension was the age range of the sample (18–55), whereas previous studies have indicated that the highest rate of hypertension exists among the elderly population [36,37]. The findings showed no significant relationship between depressive symptoms and chronic physical illnesses, which is inconsistent with previous studies [21,22,23,24,25]. This could be related to several reasons, including the onset, duration, severity, and type of treatment used for either depression or physical illnesses. In addition, some previous studies included only those with clinical depression (i.e., who met the DSM-5-TR criteria for major depression), which was different from our study, which included individuals who showed depressive symptoms according to the PHQ-9 scores.

Objective physical health indicators highlighted that the current physical health status of individuals with depressive symptoms was not optimal. For example, approximately 12% of participants experienced either bradycardia or tachycardia, and more than one-third of participants were either overweight or obese. This was supported by the abnormally high waist-to-hip ratio among both males (21.9%) and females (28.8%). Additionally, this finding was supported by the self-reported questionnaire, since approximately half of the participants reported noticeable weight gain during the previous year. Townsend and Morgan [3] explained that patients with depression may experience weight gain or weight loss during depressive episodes, which could also be considered a coping strategy for individuals during depression. It was also clear that the largest percentage of weight gained was in the form of fat, which could have explained the consistent increase and the association between the increased waist-to-hip ratio and overweight or obesity. Our findings are consistent with a study conducted in Korea, in which researchers reported a strong relationship between body mass index and depressive symptoms [40]. Additionally, consistent with our findings, Noh et al. [40] reported that patients who were underweight, overweight, or obese had a significantly greater depression severity than patients with a normal weight. Additionally, these findings are consistent with those of a study conducted in Peru, where researchers found that an increased waist-to-hip ratio was significantly associated with depressive symptoms [20]. Finally, in the current study, 10.6% of the participants had limited lung capacity because their FEV1/FVC ratio was less than 0.70, which may have increased their risk of developing restrictive or obstructive lung disease. This was supported by the 19.1% of participants who reported having a wheezing chest during the previous year, and the fact that they had significantly higher mean scores of depressive symptoms than participants who did not experience this symptom. This finding is consistent with a study conducted by Peng et al. [18], who reported a significant relationship between depression status and the FEV1/FVC ratio. Patients with depression had a significantly lower FEV1/FVC ratio than nondepressed patients, which may have increased the risk of obstructive lung disease among depressed patients [18].

The findings of the self-reported questionnaire were consistent with the vital signs (i.e., heart rate and blood pressure), anthropometric data, and lung capacity, and all of them revealed that the physical health of individuals with depressive symptoms was not optimal. For example, the percentage of individuals who complained of either chest pain, headache/migraine, or body pain during the previous year ranged from 12.2 to 43.2%. Previous studies explained that mental disorders, including depression, may have two-way relationships with physical illness and physical symptoms [3,13,14]. If depressive symptoms remain untreated for a long period of time, physical symptoms may be triggered, and vice versa [3,13,14]. These findings are consistent with those of a previous study that revealed that depressed patients had significantly higher rates of chronic pain with a greater severity than nondepressed patients, including pain in different areas of the body, such as cardiac, respiratory, musculoskeletal, and gastrointestinal pain [41]. The findings of all aspects of physical health that were examined in this study supported the last question in the survey, which was about the overall rating of health, and consistently, 28.5% of participants with depressive symptoms rated their health as either fair or poor.

Several factors in this study were significantly related to depression scores: heart rate, waist-to-hip ratio, headache/migraine, chest pain, body pain, and recent weight change. These findings are consistent with those of previous studies [20,40,41,42]. These physical signs and symptoms have a two-way relationship with depression, which means that healthcare professionals who care for depressed patients should consider assessing these physical signs and symptoms, and other physical health indicators in general. Also, healthcare professionals who take care of patients with physical diseases in general (especially those who complain of chronic pain, reported recent weight changes, or who have manifested abnormal heart rates or abnormal waist-to-hip ratios) should be screened for depression. Consequently, healthcare professionals can detect depression or physical illness earlier, which may prevent complications and the exacerbation of the client’s case.

One of the strengths of this study is the use of a large, population-based dataset from a non-Western context, which allows for the exploration of understudied health dynamics. The inclusion of culturally adapted instruments and robust statistical controls for chronic disease comorbidities further enhances the reliability of the findings. On the other hand, this study has several limitations. First, the secondary analysis may limit the choices of the operational definitions for the variables of interest because the data are already collected, and although the QBB uses highly valid and reliable tools/instruments to measure most of the variables, such as depression, lung capacity, and vital signs, the self-reported questionnaire about physical health has some limitations in the way the questions were developed. Therefore, we recommend the use of standardized tools/instruments that are highly valid and reliable to measure physical health subjectively. Second, the cross-sectional design limits the ability to infer causality between variables; although we tried to partially determine the predictors of depression scores through multiple linear regressions, we recommend conducting future studies that use longitudinal, experimental, or quasi-experimental designs to establish causality. Third, the data did not include direct questions about the diagnosis of depression and the type of treatment used (pharmacological vs. non-pharmacological treatment) for those who were diagnosed with depression. These questions are important to better understand the nature of the relationship between depression and physical health. Future research should aim to investigate how different cultural, economic, and healthcare contexts influence the link between depression and physical health, particularly in developing or rapidly transitioning societies.

## 5. Conclusions

The physical health of individuals with depressive symptoms was not optimal. Those individuals reported several comorbidities such as diabetes, hypertension, and hypercholesterolemia. Additionally, more than one-third of the participants were either overweight or obese, and a large percentage of both males and females had an abnormally high waist-to-hip ratio. Also, individuals with depressive symptoms complained of several physical symptoms including a wheezing chest, headache/migraine, body pain, and chest pain. Consequently, a large percentage of the sample reported their general physical health as being fair or poor. Several physical health symptoms and indicators were significantly related to the depression scores. This highlighted the strong association between physical health and mental health. Healthcare professionals should screen individuals for both depressive symptoms and physical symptoms. This may help in the earlier detection of mental illness and/or physical disease and thus may ensure better recovery or prognosis and prevent serious complications. A multidisciplinary healthcare team should be available in healthcare settings to treat or manage physical and depressive symptoms simultaneously. Public health policies aimed at reducing the burden of depression should take into account the close interplay between mental and physical health in adult populations.

## Figures and Tables

**Table 1 ijerph-22-00726-t001:** Sample characteristics.

Category	*n* (%)	Relationship with PHQ-9
Gender		
Female	386 (56.2)	*t*-test
Male	301 (43.8)	t = 1.73, *p* = 0.08
Education level		
High school	120 (17.5)	
Technical (diploma) degree	135 (19.6)	ANOVA
University (Bachelor) degree	366 (53.3)	F = 0.69, *p* = 0.74
Postgraduate degree	39 (5.7)	
Others	27 (3.9)	
Employment		
In paid employment	447 (65.1)	
Retired	23 (3.3)	ANOVA
Housewife	72 (10.5)	F = 0.40, *p* = 0.96
Student	81 (11.8)	
Unemployed	24 (3.5)	
Others	40 (5.8)	
Are you diagnosed with diabetes?		
Yes	69 (10.0)	*t*-test
No	618 (90.0)	t = 0.30, *p* = 0.77
Are you diagnosed with hypertension?		
Yes	86 (12.5)	*t*-test
No	601 (87.5)	t = 0.13, *p* = 0.89
Are you diagnosed with hypercholesterolemia?		
Yes	180 (26.2)	*t*-test
No	507 (73.8)	t = 0.37, *p* = 0.71

Note: PHQ-9: Patient Health Questionnaire-9. ANOVA: Analysis of Variance.

**Table 2 ijerph-22-00726-t002:** Descriptive statistics of age and physical health indicators.

Variable	Mean (SD)	Range	Pearson Correlation (r) with PHQ-9
Age	35.39 (9.82)	18–55	−0.027
Heart Rate	72.09 (12.94)	51–123	0.079 *
Systolic Blood Pressure	111.62 (13.10)	83–170	−0.023
Diastolic Blood Pressure	70.34 (10.37)	51–113	−0.005
BMI	28.81 (6.41)	17.20–48.30	0.045
Waist-to-hip Ratio	0.82 (0.10)	0.62–1.00	−0.076 *
FEV1/FVC	0.79 (0.10)	0.46–1.00	−0.021

Note: * *p* < 0.05. PHQ-9: Patient Health Questionnaire-9. BMI: body mass index. FEV1/FVC: Forced Expiratory Volume/Forced Vital Capacity.

**Table 3 ijerph-22-00726-t003:** Descriptive statistics of self-reported physical symptoms.

Variable	Category	*n* (%)	PHQ-9 Scores ± SD	Relationship with PHQ-9
Wheezing chest in the last year?	Yes	131 (19.1)	10.73 ± 4.38	*t*-test
	No	556 (80.9)	9.90 ± 3.94	t = −2.01, *p* = 0.03 *
Frequent headache or migraine in the last year?	Yes	297 (43.2)	10.76 ± 4.27	*t*-test
	No	390 (56.8)	9.76 ± 3.98	**t = −2.96** **
Pain in several areas of the body (or all over the body) in the last year?	Yes	84 (12.2)	12.57 ± 5.23	*t*-test
	No	603 (87.8)	9.88 ± 3.82	**t = −4.52** ***
Chest pain in the last year?	Yes	282 (41.0)	10.75 ± 4.42	*t*-test
	No	405 (59.0)	9.60 ± 3.70	**t = −3.58** ***
Weight change during the last year?	Approximately the same	163 (23.7)	9.25 ± 3.56	ANOVA F = 4.57 *
	Yes, gained weight	331 (48.2)	10.42 ± 4.33	
	Yes, lost weight	193 (28.1)	10.16 ± 3.91	
In general, how would you rate your health?	Excellent	125 (18.2)	8.81 ± 3.36	ANOVA F = 29.69 ***
	Good	366 (53.3)	9.61 ± 3.58	
	Fair	164 (23.9)	11.03 ± 4.03	
	Poor	32 (4.6)	15.28 ± 6.31	

Note: Bold indicates significant results after alpha adjustment for Bonferroni correction. Note: * *p* < 0.05, ** *p* < 0.01, *** *p* < 0.001. PHQ-9: Patient Health Questionnaire-9. ANOVA: Analysis of Variance.

**Table 4 ijerph-22-00726-t004:** Simultaneous multiple linear regression model: Predictors of the PHQ-9 scores.

Variable	Unstandardized B	β	t
Constant	7.983		4.554
Headache/Migraine	0.238	0.028	0.655
Body Pain	2.378	0.199	4.671 **
Chest Pain	0.747	0.089	2.146 *
Heart Rate	0.031	0.099	2.448 *
Waist-to-hip Ratio	−1.804	−0.042	−1.037
Weight Change, Increased (Dummy Variable)	0.977	0.117	2.351 *
Weight Change, Decreased (Dummy Variable)	0.893	0.095	1.903

Note: * *p* < 0.05, ** *p* < 0.001.

## Data Availability

A data request can be made by contacting the Qatar Biobank team.

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
