# Peer review of "Physical Health Among Adults with Depressive Symptoms in Qatar: Findings from Qatar Biobank Population-Based Study"

_ijerph, 2025, doi:10.3390/ijerph22050726_

Round 1
Reviewer 1 Report
Comments and Suggestions for Authors
Dear Authors,
While reviewing the manuscript, I have provided some suggestions that I believe could further strengthen your work. My aim is to assist in making the methodology, discussion, and conclusions clearer and more effective, and to help improve the quality of the manuscript. I hope the comments I have provided will be helpful and constructive in developing your paper.
Introduction
The introduction does not explicitly state a clear hypothesis. The study examines the relationship between physical health indicators and depressive symptoms, but it does not present a hypothesis regarding the direction of this relationship. A hypothesis could clarify the link between depression and physical health, making a stronger connection with the study's objectives. This would help the research adopt a more testable and targeted structure. Therefore, a hypothesis should be added, and if possible, the direction of the hypothesis should be specified.
Additionally, more references to previous studies, particularly large-scale meta-analyses or systematic reviews examining the relationship between depression and physical health, could be included. Furthermore, more information could be provided about how depression affects these physical health indicators (e.g., the physiological mechanisms through which depression leads to these outcomes). Expanding on the literature could further strengthen the context of the study.
Finally, a more technical language could be used, and when emphasizing the importance of the research, more attention could be drawn to the societal context of the impact of depression on physical health. This would be important for engaging the reader's interest.
Materials and Methods
Demographic, health history, and lifestyle information of the participants could be provided at the beginning of the methods section.
It would be beneficial to provide more details about the frequency of physical health measurements and the data collection process. For example, when and under what conditions were the participants' physical health data collected? This would add an additional layer of detail that could increase the reliability of the results. Furthermore, if the data collection process was carried out by experts (e.g., nurses and clinical staff), it could improve the accuracy of the physical health data. This information should be added.
Results
This section is generally well-structured within the study.
Discussion
The study’s findings are clearly presented. The section comparing the findings with the literature and connecting them with theory could be expanded upon.
Conclusion
The conclusion is generally clear and concise. However, more details could be provided on how the results could have a specific impact and how the findings might be applied in practice. Currently, the conclusion focuses mainly on the findings themselves, and more detailed suggestions on how these findings could be used in clinical practice or policy development would be helpful.
It might also be beneficial to expand the conclusions to a broader perspective. For instance, a few sentences could be added about the potential effects of depression not only on individual health but also on public health and health policy.
Reviewer 2 Report
Comments and Suggestions for Authors
Respected Authors,
The manuscript is well written and presented neatly. Good work!
I have one suggestion. Under discussion or conclusion, you could have mentioned some preliminary recommendations on how to improve the physical health of people suffering from depressive symptoms. Or suggest to conduct a separate study.
Regards.
Reviewer 3 Report
Comments and Suggestions for Authors
dear authors,
this manuscript was interesting because the author wants to analyze physical health among people with depressive symptoms in Qatar: Findings from Qatar Biobank population-based study.
However, you include respondents with degenerative diseases (i.e., diabetes, hypertension, and hypercholesterolemia) that with a diagnosis of those NCDs already add some stress to the respondent that may link to their depressive symptoms. However, the author still can improve in the discussion part about this specific respondent's condition that can affect the outcome and how you deal with that issue.
Please provide more relevant references that can support your manuscript so the novelty of this study can be clearer than before.
Is there any possibility that the respondents may had diagnosed with more than 1 condition? if yes, how do the authors treat those sample?
There is no table that can support the idea about how the healthy respondents compare to those with chronic condition related to the outcome. Please explain about it and add some reason why author did not add that information.
Reviewer 4 Report
Comments and Suggestions for Authors
Dear Editors and Authors,
Thank you for the opportunity to review the manuscript titled “Physical health among people with depressive symptoms in Qatar: Findings from Qatar Biobank population-based study.” This paper presents an original cross-sectional study investigating the physical health status of adult individuals with depressive symptoms in Qatar and examining the relationship between physical health indicators and depressive symptoms.
The researchers utilized data from the Qatar Biobank, which collected information from more than 24,000 participants in 2012. A sample of 687 individuals aged 18–55 years was extracted, including participants with at least mild depressive symptoms who had lived in Qatar for at least three years. The Patient Health Questionnaire-9 (PHQ-9) was employed, and sociodemographic data along with multiple physical health indicators were collected. The results highlight that several factors were significantly related to depression scores: heart rate, waist-to-hip ratio, headache/migraine, chest pain, body pain, and recent weight change. These findings provide important insights for healthcare professionals.
While the study is interesting due to its topic and its focus on a non-American and non-European sample, I have many concerns about the manuscript. Numerous changes and clarifications, particularly in conceptual and methodological aspects, are necessary to improve the quality of the paper before it can be accepted. Below, I provide comments on the manuscript section by section, not in order of importance. The title accurately reflects the main topic of the paper, though “among adults” might be more appropriate than “among people.”
The title accurately reflects the main topic of the paper, though “among adults” might be more appropriate than “among people.” The abstract provides a comprehensive overview of the manuscript, and the keywords are adequate.
The introduction is a weak part of the manuscript and resembles more the introduction of a short report than that of a full research article.
I suggest that the authors begin the introduction with a paragraph discussing the concept of health and its various dimensions—physical, psychological, and social—as traditionally conceptualized in the biopsychosocial model. Framing the study within this holistic perspective would strengthen the theoretical foundation of the paper and help contextualize the relationship between depressive symptoms and physical health indicators. It would also highlight the importance of considering mental health not in isolation, but as an integral component of overall well-being.
Relevant literature supporting this approach includes Santrock, J. W. (2007). A Topical Approach to Human Life-span Development (3rd ed.). McGraw-Hill; and Frankel, R. M., Quill, T. E., & McDaniel, S. H. (Eds.) (2003). The Biopsychosocial Approach: Past, Present, Future. University of Rochester Press.
The introduction addresses depression only in general terms and does not specifically refer to its manifestation or implications in adulthood.
Although it incorporates concepts previously used in the literature, I encourage the authors to frame their introduction more clearly in relation to adult age, particularly considering the psychosocial and physiological changes that occur during this stage of life. Additionally, the literature review is rather limited; I recommend including more relevant and up-to-date studies that align more closely with the aims of the current research, and providing more detail on the findings and methodologies of the studies cited.
As a reader, I would also expect a more comprehensive description of the diagnostic criteria for Major Depressive Disorder, as defined in the DSM-5-TR (2022), to support the conceptual and clinical framework of the study. Regarding lines 45–47, the authors briefly mention that the relationship between depression and physical illness is bidirectional, but this important concept requires further elaboration, supported by empirical evidence.
Finally, the rationale for the study should be more clearly articulated and better grounded in existing literature, especially concerning the specific characteristics of the adult population in Qatar.
Why is this study important? What gap in the literature does it aim to fill? Clarifying these points would greatly enhance the strength and relevance of the introduction.
The methods employed are rigorous, well-explained, and appropriate for the study’s aims. Sufficient information is provided to allow a competent researcher to replicate both the survey and the statistical analyses, as the procedures and instruments are clearly described. The statistical methods used are suitable for addressing the research objectives. However, the scientific soundness of the study would be improved by reporting Cronbach’s alpha coefficient for the PHQ-9 within the current sample. Additionally, it is important to clarify whether a version of the PHQ-9 validated for use in Qatar was employed, and if not, to explain the process by which the questionnaire was translated into Arabic and culturally adapted.
The results are presented clearly and in an appropriate format, with tables effectively summarizing essential data that could not be easily integrated into the main text, thereby facilitating interpretation. All plausible interpretations of the data appear to have been considered, and no alternative hypotheses consistent with the available data seem to have been overlooked.
However, the Results section requires improvement in several areas. Regarding the analyses shown in Table 3, a series of t-tests were conducted. It is not clear whether the authors applied a correction for multiple comparisons, such as the Bonferroni correction. This point should be addressed explicitly, as failing to correct for multiple testing can increase the risk of Type I error.
Furthermore, concerning the multiple regression model, the type of regression should be clearly specified—was it a hierarchical regression, and if so, what variables were entered in each step? I recommend that the authors present the regression results in a dedicated table including the Constant, unstandardized Beta, standardized Beta, Standard Error (SE), R Square, and Delta R Square values to enhance transparency and interpretability.
The discussion is generally well-written and easy to follow, although there is room for improvement (see comments below). While the references cited are recent and relevant, both their number and scientific depth are somewhat limited given the complexity of the topic. The reference list could be strengthened by incorporating the suggestions made earlier regarding the introduction and literature review.
In addition, several aspects of the discussion would benefit from further development. For instance, could the authors identify and elaborate on some of the strengths of their study, beyond simply listing its limitations? Moreover, it would be helpful if the authors could suggest possible directions for future research in this field, particularly concerning adult populations in non-Western contexts.
Best regards!
Round 2
Reviewer 4 Report
Comments and Suggestions for Authors
I am pleased to see the improvement of this manuscript. It is now a well-written study presenting compelling data. Therefore, I recommend that the Editor(s) consider this paper for publication in the prestigious journal IJERPH.